# Flow Pattern Identification of Oil–Water Two-Phase Flow Based on SVM Using Ultrasonic Testing Method

**DOI:** 10.3390/s22166128

**Published:** 2022-08-16

**Authors:** Qian Su, Jie Li, Zhenxing Liu

**Affiliations:** 1School of Information Science and Engineering, Wuhan University of Science and Technology, Wuhan 430081, China; 2Engineering Research Center of Metallurgical Automation and Measurement Technology, Wuhan University of Science and Technology, Wuhan 430081, China

**Keywords:** oil–water two-phase flow, ultrasonic attenuation, reflected echo, flow pattern recognition, SVM

## Abstract

A flow pattern identification method combining ultrasonic transmission attenuation with an ultrasonic reflection echo is proposed for oil–water two-phase flow in horizontal pipelines. Based on the finite element method, two-dimensional geometric simulation models of typical oil–water two-phase flow patterns are established, using multiphysics coupling simulation technology. An ultrasonic transducer test system of a horizontal pipeline with an inner 50 mm diameter was built, and flow pattern simulation experiments of oil–water two-phase flow were carried out in the tested field area. The simulation results show that the ultrasonic attenuation coefficient is extracted to identify the W/O&O/W dispersion flow using the ultrasonic transmission attenuation method, and the identification accuracy is 100%. By comparison, using the ultrasonic reflection echo method, the echo duration is extracted as an input feature vector of support vector machine (SVM), and the identification accuracy of the stratified flow and dispersed flow is 95.45%. It was proven that the method of the ultrasonic transmission attenuation principle combined with the ultrasonic reflection echo principle can identify oil–water two-phase flow patterns accurately and effectively, which provides a theoretical basis for the flow pattern identification of liquid–liquid multiphase flow.

## 1. Introduction

Oil–water two-phase flow in horizontal pipelines exists widely in oil production and transportation. The flow pattern of oil–water two-phase flow in horizontal wellbores is complex and changeable, which makes it difficult to accurately measure flow parameters of oil–water two-phase flow. Accurate identification of the two-phase flow pattern and the in-depth study of flow pattern conversion comprise the basic work required for the study of heat transfer and pressure drop characteristics under given flow conditions and the measurement of the process parameters of multiphase flow. The scientific description of the flow pattern characteristics of horizontal oil–water two-phase flow has important academic value and practical significance for solving critical technical problems in the petroleum industry [1]. Due to different factors such as flow rate and phase-separated holdup, the oil–water two-phase flow forms various flow patterns in the mixed flow. Rather than single-phase flow, two-phase flow has the characteristics of nonlinearity and interphase slip [2], which contribute to certain challenges to flow pattern identification and flow pattern conversion detection.

Nädler [3] studied the classification and phase conversion of oil–water two-phase flow emulsion, and seven flow patterns, namely, stratified flow of separated phases, stratified flow with mixing at interface, unstable water-in-oil emulsion (W-O), layers of water-in-oil dispersion (W-in-O) and water, layers of dispersions (W-in-O, O-in-W) and water, oil-in-water dispersion (O-in-W) above a water layer, and unstable oil-in-water emulsion (O-W) were observed in the horizontal pipeline. The occurrence of phase inversion and emulsification type were determined by measuring the electrical conductance of the oil–water flow mixture in an on-line conductance cell. Brauner [4] used the principle of minimum total system energy to predict the interface shape in the stratified flow of a general two-phase system and constructed a model algorithm based on interface curvature to predict flow pattern. Angeli [5] studied flow structure in horizontal oil–water two-phase flow using high speed videoscope and high frequency impedance probes. A total of seven flow patterns were obtained, namely, stratified wavy (SW), three layers (3L), stratified mixed/oil (SM/oil), phase continuity boundaries, stratified wavy/drops (SWD), stratified mixed/water (SM/water), and mixed (M). Moreover, experimental flow pattern maps of oil–water two-phase flow in acrylic resin and stainless steel tube pipelines are substantially different. Lu [6] used a high-speed camera and the conductivity probe method to study oil–water two-phase flow patterns and boundary characteristics in an injecting pipeline for a mix of oil and water and the two fluids separately. The flow patterns of oil–water two-phase flow in a small annular channel with an inner diameter of 25.4 mm and a large annular channel with a 40 mm inner diameter were obtained. The experimental results showed that a dispersed flow pattern is easier to form by mixing oil and water phases in the pipeline, and the smaller the pipe diameter, the easier the oil–water dispersed flow pattern forms. Huang [7] used electrical resistance tomography technology of 16 electrodes in a single section and a pressure sensor system to conduct an experimental study on flow pattern characteristics of oil–water two-phase flow in a horizontal pipeline with a 50 mm inner diameter. The conductivity distribution in the pipeline section was reconstructed using the projection algorithm, by which visualized measurements of oil and water phase distribution and the conversion boundary between flow patterns were obtained.

Multiphase flow is affected by many factors, such as the experimental environment, temperature, pressure, phase separation flow rate, pipe size, and fluid viscosity. Oil–water two-phase flow can be mainly divided into three categories, namely, stratified flow, dispersion flow, and dispersed flow, which can be further subdivided into six typical flow patterns, namely, stratified flow (ST), stratified flow with a mixed interface (ST&MI), dispersion of oil in water and water (O/W&W), dispersion of water in oil and oil in water (W/O&O/W), oil in water emulsion (O/W), and water in oil emulsion (W/O). The six flow patterns are shown in Figure 1. Trallero [8] derived six experimental flow patterns according to the phase separation velocity of oil–water two-phase flow in a horizontal pipeline, as shown in Figure 2.

Multiphase flow measurement technology is based on the use of high-speed cameras [9], conductivity probes [10], particle image velocimetry [11], electrical capacitance tomography [12], optical probes [13], ultrasonic methods [14], etc. Among these, technologies such as high-speed cameras, optical probes, and particle image velocity measurements are restricted to a tested pipeline having high transmittance. Intrusive conductivity probes can disturb multiphase flow, whereas resistance tomography techniques are not suitable for measuring flow patterns of oil continuous phase. The ultrasonic method can be used in laboratories and complex industrial environments because of its lack of contact, independence of conductivity and light transmittance, and the fact that sensors will not be corroded or worn by fluids.

In this study, the feasibility of identifying oil–water two-phase flow patterns based on the ultrasonic transmission attenuation method combined with the ultrasonic reflection echo method was verified. Finite element simulation models of oil–water two-phase flow, including stratified flow, dispersion flow, and uniformly distributed dispersed flow, were built using COMSOL Multiphysics^®^ (Version 5.6, Stockholm, Switzerland). Based on the ultrasonic propagation mechanism and characteristics analysis, an ultrasonic transducer transceiver system was established for the oil–water two-phase flow test. The ultrasonic attenuation signal and reflected echo signal were extracted as ultrasonic parameters to identify typical flow patterns of oil–water two-phase flow. Results show that the ultrasonic attenuation coefficient extracted by the ultrasonic transmission attenuation method can effectively identify the W/O&O/W flow pattern, and echo duration extracted by the reflection echo method as the input feature vector of SVM can accurately identify stratified flow and dispersed flow.

## 2. Fundamentals of Acoustic Testing

Acoustic testing methods mainly include the attenuation method [15], reflection method [16], and Doppler method [17]. A change in the oil–water two-phase flow pattern will cause amplitude attenuation and phase change of the received sound pressure signal [18]. Due to different acoustic properties propagating in oil phase and water phase, transmission and reflection properties of acoustic waves in the oil–water two-phase flow can present the relevant information of the change in the oil–water structure, namely, the flow pattern. A schematic diagram of ultrasonic attenuation measurement is shown in Figure 3. After ultrasonic absorption attenuation, scattering attenuation, and diffusion attenuation mechanisms in oil–water two-phase flow, the ultrasonic wave emitted from the ultrasonic transmitter reaches the ultrasonic receiver [19]. The ultrasonic attenuation coefficient is:(1)$$\alpha =-\frac{\ln(P_{2}/P_{1})}{l}$$
where $P_{2}$ and $P_{1}$ are the sound pressures at the ultrasonic transmitter and receiver, respectively. *l* is the distance between the transmitter and receiver.

The principle of the ultrasonic reflection test method is based on different acoustic characteristics such as the acoustic impedance of two fluids. When the ultrasonic wave is incident from one medium to another medium with different acoustic impedance at the interface, part of the ultrasonic energy is transmitted to another medium, and the other part of the energy is reflected back to the original medium. Ultrasonic transmittance and reflectance are related to the incident angle and the acoustic impedance of the media on the interface. When the ultrasonic wave is vertically incident to the interface of two different media, the transmission coefficient and reflection coefficient of sound pressure are, respectively:(2)$$\tau_{P}=\frac{2Z_{2}}{Z_{1}+Z_{2}}=\frac{2}{1+\varepsilon }$$
(3)$$\gamma_{P}=\frac{Z_{2}-Z_{1}}{Z_{1}+Z_{2}}=\frac{1-\varepsilon }{1+\varepsilon }$$
where $\tau_{P}$ is the sound pressure transmission coefficient, and $\gamma_{P}$ is the sound pressure reflection coefficient. $Z_{1}$ and $Z_{2}$ are, respectively, acoustic impedances of the two media. $\varepsilon =\frac{Z_{1}}{Z_{2}}$ is the acoustic impedance ratio.

## 3. Modeling and Simulation of Oil–Water Two-Phase Flow

Based on the finite element method, a two-dimensional geometric model of the measured area was established by COMSOL Multiphysics^®^ simulation software. The size of the ultrasonic transducer was 0.6 × 9 mm, and the inner diameter of the pipe was 50 mm. The transmitting end of the ultrasonic transducer was installed directly below the pipeline, and the receiving end was located directly above the pipeline, as shown in Figure 3. The bottom ultrasonic transducer was in a spontaneous and self-receiving mode, and the ultrasonic center emission frequency was 1 MHz. The distribution of dispersed phase droplets was simulated by circles with diameters of 1, 1.5, 2, 2.5, and 3 mm. 

The ultrasonic center emission frequency was 1 MHz. The ultrasound velocity in oil and water fluids was 1420 and 1448 m/s, respectively, at an environment temperature of 293.15 K. The density of oil and water was 850 and 1000 kg/m^3^, respectively.

The transmitter was set as the structure-acoustic coupling method. The governing equation of the simulation model was set inside the multi-physical field of transient pressure acoustics using COMSOL Multiphysics^®^. The oil phase and water phase regions were meshed separately with free triangular elements. According to the wavelength of the ultrasound in the oil phase and water phase, the size of the mesh elements was 1/6 of the wavelength of the ultrasound so as to balance the simulation precision and computational cost. The set boundary conditions for the relevant physical parameters of the two-phase fluid are listed in the Table 1.

### 3.1. Modeling and Simulation of Stratified Flow

The mixed apparent velocity is an important factor affecting the formation of the flow pattern of oil–water two-phase flow [8]. At low mixed apparent velocity, the model of laminar flow resistance plays a primary role and the stratified interface is obvious. Under this condition, the flow pattern of stratified flow is formed. At a given mixed apparent flow velocity, the interface height decreases with the increase in oil phase fraction, and the oil–water interface curvature also changes with the oil phase fraction [20]. The model of ST flow is simulated in Figure 4. ST flow patterns with different oil fractions are represented by adjusting the position and curvature of the oil–water interface. Typical simulation results of sound pressure distribution of some ST flow patterns are shown in Figure 5, where the oil fractions are 20%, 40%, and 60%. In order to obtain sound pressure distribution in the whole pipeline, the simulation results take the moment about 3.4 × 10^−5^ s when ultrasonic waves reach the top of the pipeline. The parameter settings of simulation experiments described in the following section remained the same.

With the increase in superficial mixing velocity, the model of laminar flow resistance gradually fails, which results in the entrainment of one fluid into another, forming a mixed stratified flow at the phase interface. The relationship between superficial mixing velocity and oil phase fraction with the change in droplet size was presented in the study of Markides [21]. When the oil phase fraction is constant, the droplet size increases with the increase in superficial mixing velocity. When superficial mixing velocity is constant, the droplet size increases with the increase in oil phase fraction. The model of ST&MI flow is simulated in Figure 6. ST&MI flow patterns with different oil fractions were modeled by changing the interface position and curvature, and adjusting the droplet size according to the relationship between the phase fraction and the superficial mixing velocity. Typical simulation results of sound pressure distribution of some ST&MI flow patterns are shown in Figure 7, where oil fractions are 20%, 40%, and 60%.

With the increase in water fraction, the disturbance of water to oil increases, and the layer of dispersed oil droplets in the upper water becomes thicker, gradually forming a stratified flow of oil in water and water. The model of O/W&W flow is simulated in Figure 8. O/W&W flow patterns with different oil fractions are established by changing the thickness of the dispersed oil droplet layer and the concentration of dispersed oil droplets. Typical simulation results of some O/W&W flow patterns are shown in Figure 9, where oil fractions are 10%, 15%, and 20%.

### 3.2. Modeling and Simulation of Dispersion Flow

Angeli [9] studied the droplet size and concentration in bicontinuous dispersed flow and found that the farther away from the oil–water interface, the smaller the droplet size. The model of W/O&O/W flow is simulated in Figure 10. Typical simulation results of some W/O&O/W flow patterns are shown in Figure 11, where oil fractions are 40%, 50%, and 60%.

### 3.3. Modeling and Simulation of Dispersed Flow

When superficial mixing velocity is high, oil and water fluids are fully mixed, and form characteristics of oil–water two-phase dispersed flow. The dispersed flow can be divided into oil-in-water-flow with water-based continuous phase and water-in-oil flow with oil-based continuous phase. When the water fraction of mixed fluids at the inlet is high, the oil phase is dispersed in the water phase, whereas when the oil phase fraction is high, the water phase breaks up and disperses in the oil continuous phase. The distribution of dispersed droplets from the bottom to the top of the pipeline has a certain concentration distribution gradient. With the increase in the superficial mixing velocity, the distribution of the dispersed droplets gradually becomes uniform. The phase fraction is changed by adjusting the density of dispersed droplets. The model of O/W flow is simulated in Figure 12, and typical simulation results of some O/W flow patterns are shown in Figure 13, where oil fractions are 20%, 40%, and 60%. The model of W/O flow is simulated in Figure 14, and typical simulation results of some W/O flow patterns are shown in Figure 15, where oil fractions are 70%, 80%, and 90%.

## 4. Characteristics Analysis of Ultrasonic Signal

### 4.1. Ultrasonic Attenuation Method

Based on the ultrasonic attenuation principle, the ultrasonic attenuation coefficients were extracted for each flow pattern analysis of oil–water two-phase flow. The ultrasonic attenuation coefficient of each flow pattern simulation result was calculated by Equation (1), as shown in Figure 16. The ultrasonic attenuation coefficient of W/O&O/W flow ranges from 14 to 19 dB/m in the oil fraction between 40% and 60%, whereas the ultrasonic attenuation coefficient of other flow patterns is in the range of 9 to 14 dB/m in the oil fraction between 10% and 90%. The results show that W/O&O/W flow pattern can be distinguished from the six flow patterns by the ultrasonic attenuation method.

### 4.2. Ultrasonic Reflection Method

Based on the ultrasonic reflection principle, the waveform of the echo signal at the bottom ultrasonic transducer under each oil–water two-phase flow pattern was analyzed. It was found that the amplitude and frequency of reflected echoes are closely related to the size, shape, and position of the phase interface. Figure 17 and Figure 18 show the time-domain distribution of absolute sound pressure by the bottom transducer with the “self-transmitting and self-receiving” [22] mode, under stratified flow and dispersed flow with different oil fractions. The ultrasonic emission was simulated for a period of about 1.5 × 10^−5^ s, so the reflected echo signals after 1.5 × 10^−5^ s were analyzed.

From Figure 17a,b, it can be seen that the echo duration is about 1 × 10^−5^ s, and the echo intensity is relatively high. Combined with the instantaneous sound pressure distribution at 3.4 × 10^−5^ s of the ST flow and ST&MI flow patterns shown in Figure 5 and Figure 7, reflected echoes are relatively focused, and most of the echoes’ energy is reflected vertically back to the transducer. This is mainly caused by the different acoustic impedance of the two fluids and the relatively smooth oil–water interface. From the reflected echo signals of oil fractions of 20%, 40%, and 60%, it was found that the transit time of the reflected echoes to the bottom receiver decreases with the increase in oil fraction. 

From Figure 17c of the O/W&W flow pattern, the reflected echoes distribute at the far end after the emission pulse and the echo intensity is relatively low. Oil droplets of O/W&W flow are mainly distributed in the upper layer of the pipeline, and the reflected acoustic signal seriously attenuates over long distances, so the echo intensity in O/W&W flow is lower than that in other flow patterns. The echo duration of O/W&W flow is longer than that of stratified flow including ST and ST&MI patterns.

Figure 18 shows the reflected ultrasound of O/W and W/O flow patterns with different oil fractions are relatively diffused. The echo duration of O/W flow lasts from 1.5 × 10^−5^ to 6 × 10^−5^ s in the time domain, and the echo duration of W/O flow starts at 1.5 × 10^−5^ s, whereas the echo cut-off time changes with the water fraction. The echo intensity of the O/W flow pattern is higher than that of the W/O flow pattern, and the echo intensity of both patterns has a positive correlation with the dispersed phase fraction. Combined with the distribution of absolute sound pressure in Figure 13 and Figure 15, the propagation direction of the ultrasonic wave forms a certain angle with the surface of the dispersed phase, which contributes to a serious scattering in dispersed flow including O/W and W/O flow patterns. The distance between dispersed droplets and the sensor in the O/W&W pattern is greater than that in O/W and W/O flow patterns, so the echo intensity of dispersed flow is low due to long-distance propagation attenuation.

## 5. Flow Pattern Identification Based on SVM

### 5.1. Support Vector Machine for Identification

Support Vector Machine (SVM) is a supervised machine learning method that is widely used for flow pattern identification. By relaxation vector and kernel technology, SVM is suitable for small and non-linear samples, compared with other machine learning algorithms having the same complexity [23]. The kernel function in SVM can deal with high-dimensional samples. In addition, it shows unique advantages, such as fast response and strong robustness to noisy data, in pattern identification with the existing small samples. The learning objective of SVM is to find an optimal decision hyper-plane in n-dimensional space and ensure the sample points in the training set are as far away from the hyper-plane as possible. $x_{i}$ in a supposed dataset $\left\{(x_{1},y_{1}),(x_{2},y_{2}),...,(x_{i},y_{i})\right\}$ is the sample feature data, and $y_{i}$ is the sample category label, where $y_{i}\in \{+1,-1\}$. If $y_{i}=1$, the sample is the first category. If $y_{i}=-1$, the sample is the second category, as seen in Figure 19. The description equation of the hyper-plane is:(4)$$W^{T}x+b=0$$

If $y_{i}=W^{T}x+b=1$, the sample becomes the support vector. If is $y_{i}$ greater or less than 1, the samples belong to the first category and the second category, respectively, as shown in Figure 20. The decision function of the SVM binary classifier is:(5)$$f(x)={\left(\sum_{i=1}^{n}\alpha_{i}{}^{*}y_{i}(x_{i},x)\right)}^{T}+b$$

### 5.2. Feature Extraction and Flow Pattern Identification

According to the characteristic analysis of the sound pressure signal above, W/O&O/W flow pattern can be identified firstly by the ultrasonic transmission attenuation method, and then stratified flow and dispersed flow patterns can be identified according to the ultrasonic reflection test method. The 132 groups of oil–water two-phase flow data in the simulation modeling were distinguished according to the flow pattern identification method shown in Figure 20. From Figure 16, the ultrasonic attenuation coefficients of the W/O&O/W flow pattern in different oil fraction ranges are all greater than 14 dB/m, whereas the ultrasonic attenuation coefficients of other flow patterns including stratified and dispersed patterns in different oil fraction ranges are less than 14 dB/m. The W/O&O/W flow pattern can be identified by the ultrasonic attenuation coefficient and the identification accuracy is 100%.

For the stratified flow and dispersed flow patterns, the reflected echo signals of the bottom receiver lasting 1.5 × 10^−6^ and 6.0 × 10^−6^ s were analyzed using the ultrasonic reflection method. Echo duration plays a predominant role in assessing classification accuracy, in terms of the six feature parameters of echo duration, echo start time, echo disappearance time, and average, peak, and variance of the absolute sound pressure signal, by evaluating the effectiveness of each feature in the flow pattern classification. Based on analyzing time-domain features that can express the phase distribution of flow patterns and waveform features that can reflect the intensity changes of reflected echoes, echo duration is extracted as the input feature vector of SVM for classification training, and 111 groups of samples are distributed in Figure 21. 

Classification accuracy would be affected by kernel functions of SVM classifier. SVM classification results of different kernel functions are shown in Table 2. The identification accuracy of the polynomial kernel function is 72.73%, whereas that of others, including linear, RBF, and sigmoid kernel functions, is 95.45%. Compared with linear and sigmoid kernel functions, the RBF kernel function has less computation time and higher efficiency. Based on the support vector machine algorithm, the SVM classifier with the RBF kernel function was selected to comprehensively satisfy the real-time and accuracy requirements of the identification system.

### 5.3. Experimental Results and Analysis

Using the ultrasonic attenuation method, the ultrasonic attenuation coefficient from the transmitting end to the receiving end was extracted to identify the dispersion flow (W/O&O/W flow pattern), and the identification accuracy was 100%. For stratified flow and dispersed flow patterns with ultrasonic attenuation coefficients less than 14 dB/m, the reflected echo signals by the bottom receiver in the time range from 1.5 × 10^−6^ to 6.0 × 10^−6^ s were analyzed. The echo duration was extracted as the input feature vector of the SVM classifier, and the identification accuracy of ST, ST&MI, O/W&W, O/W, and W/O flow patterns was 95.45%. The results show that the ultrasonic attenuation method combined with the reflection method based on SVM is effective and accurate for oil–water two-phase flow pattern identification, and completely covers flow patterns under certain experimental conditions.

The ultrasonic reflected echo signal from the theoretical analysis is basically consistent with the reflected echo signal obtained from the simulation. However, there are still discrepancies between the O/W&W flow pattern and the W/O flow pattern with high oil fractions. Water droplets are small and dispersed when the oil fraction increases over 90% in the W/O flow pattern, and the layer of dispersed oil droplets thickens with an oil fraction over 25% in the O/W&W flow pattern. Therefore, the reflected echo signals with low echo intensity represent diffusion in both flow patterns with high oil fraction, which reduces identification accuracy.

## 6. Conclusions

(1)Based on the oil–water two-phase flow dynamics research, finite element simulation models of oil–water two-phase flow, including stratified flow, dispersion flow, and uniformly dispersed flow, were established using COMSOL Multiphysics^®^. By analyzing the ultrasound transmission theories, an ultrasonic testing simulation system based on the finite element method is proposed, based on the principle of ultrasonic transmission attenuation combined with ultrasonic reflection echo. The multiphase flow simulation platform established in this paper provides a theoretical basis for the practical experimental study of multiphase flow. Scientific description of the flow pattern characteristics of horizontal oil–water two-phase flow has important academic value and practical significance for solving critical technical problems in the petroleum industry.(2)The ultrasonic attenuation coefficient was extracted to identify the W/O&O/W dispersion flow using the ultrasonic transmission attenuation method, and the identification accuracy was 100%. In addition, echo duration and echo intensity were applied by the ultrasonic reflection echo method, as an input feature vector of support vector machine (SVM), and the identification accuracy of stratified flow and dispersed flow patterns was 95.45%. A low-cost, non-intrusive, non-interference ultrasonic testing method with SVM was achieved in this work for oil–water two-phase flow pattern identification, completely covering the flow patterns of stratified flow, dispersed flow, and dispersion flow. The method provided a basic foundation for flow pattern identification of liquid–liquid multiphase flow, effectively solving the problems of the existing ultrasonic technique applied to the multiphase flow, namely, its low accuracy and the inability to identify the flow. The measurement method proposed in this paper is helpful to promote the application of ultrasonic technology without intrusion or disturbance in the field of multiphase flow pattern identification.(3)The ultrasonic attenuation method combined with the ultrasonic reflection method can distinguish the stratified and dispersed flows. However, due to the high characteristic similarity between partial flow patterns, the identification accuracy of the specifically subdividable flow patterns needs to be further improved. For example, compared with the ST flow pattern, there are unevenly distributed droplets, known as entrained droplets, at the oil–water interface of the ST&MI flow pattern. The echo intensity received by the ultrasonic transducer from the oil–water stratified interface is much larger than that from droplets. Therefore, the information of entrained droplets at the oil–water interface cannot be completely received by the ultrasonic transducer. Dispersed droplets are distributed in the continuous phase in both O/W and W/O flow patterns, and the phase fraction of the two flow patterns has overlapping areas of phase inversion in the oil–water two-phase flow map. Therefore, further study is expected to be undertaken to identify stratified and dispersed flow patterns using ultrasonic measurement technology.

## Figures and Tables

**Figure 1 sensors-22-06128-f001:**
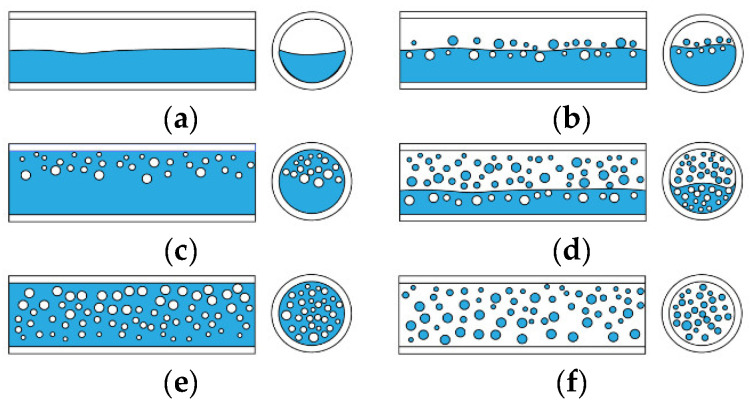
Oil–water two-phase flow pattern: (**a**) ST; (**b**) ST&MI; (**c**) O/W&W; (**d**) W/O&O/W; (**e**) O/W; (**f**) W/O.

**Figure 2 sensors-22-06128-f002:**
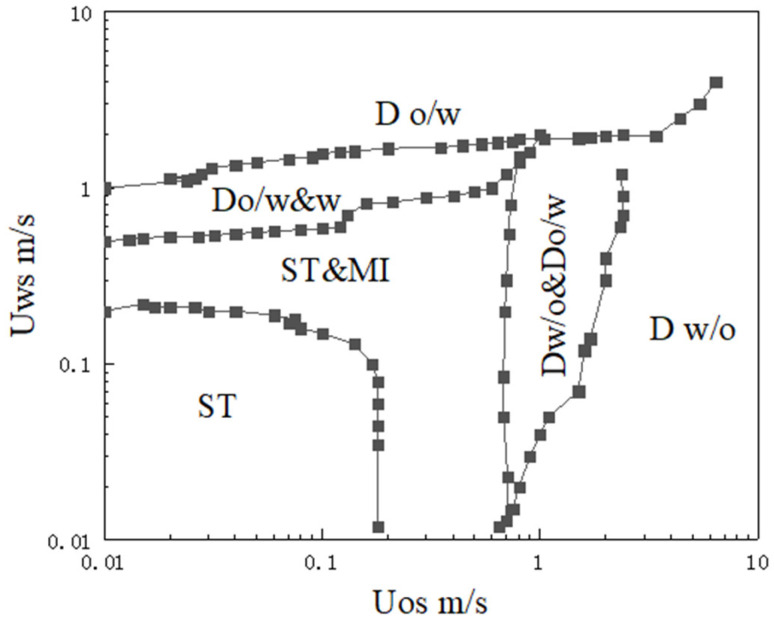
Oil–water two-phase flow experiment map.

**Figure 3 sensors-22-06128-f003:**
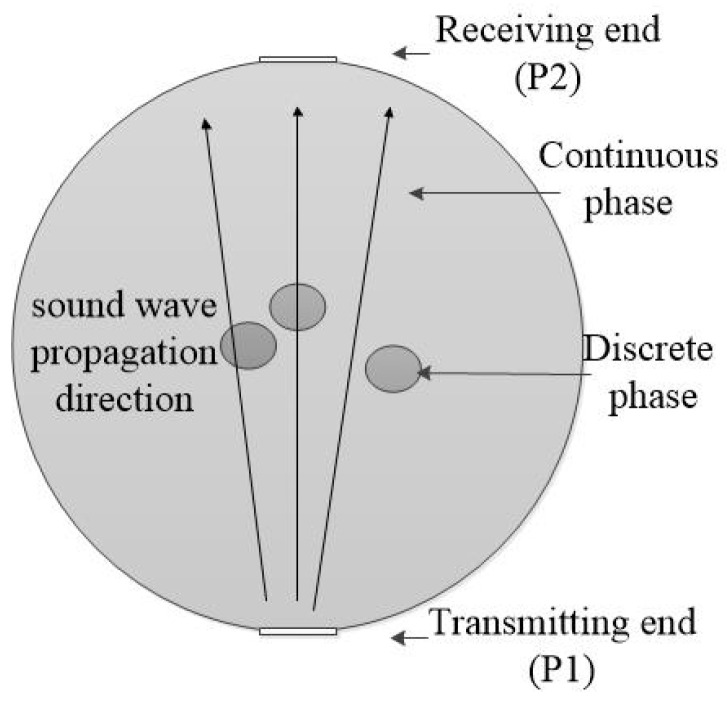
Schematic diagram of ultrasonic attenuation measurement.

**Figure 4 sensors-22-06128-f004:**
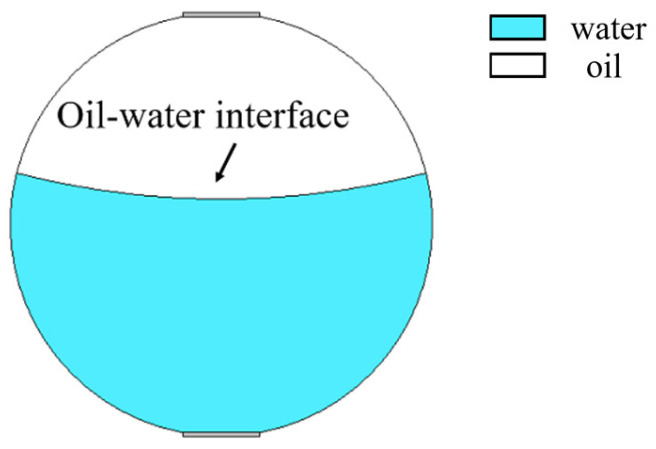
Simulation model of ST flow pattern.

**Figure 5 sensors-22-06128-f005:**
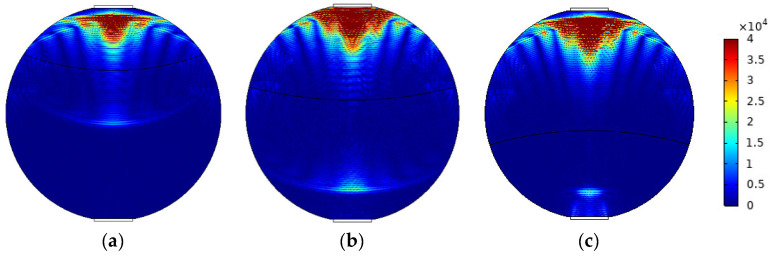
Sound pressure distribution of ST flow pattern: (**a**) oil fraction 20%; (**b**) oil fraction 40%; (**c**) oil fraction 60%.

**Figure 6 sensors-22-06128-f006:**
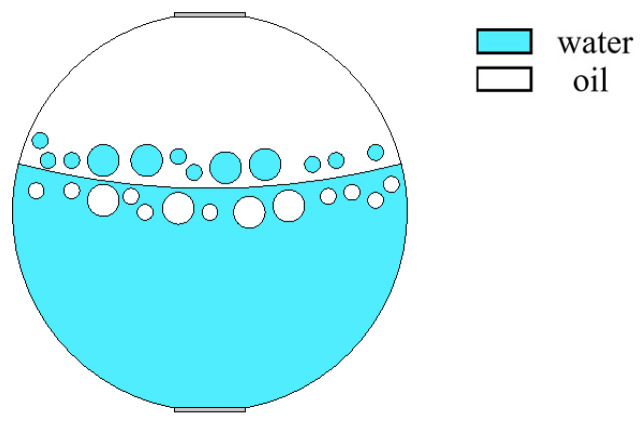
Simulation model of ST&MI flow pattern.

**Figure 7 sensors-22-06128-f007:**
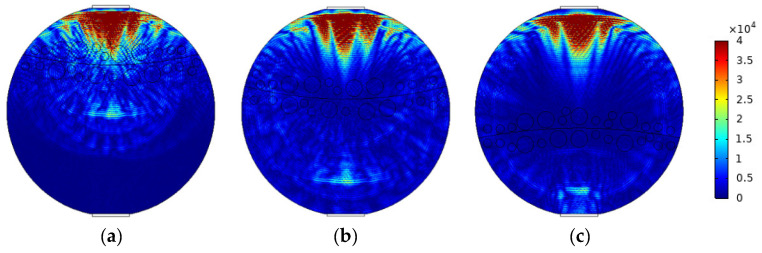
Sound pressure distribution of ST&MI flow pattern: (**a**) oil fraction 20%; (**b**) oil fraction 40%; (**c**) oil fraction 60%.

**Figure 8 sensors-22-06128-f008:**
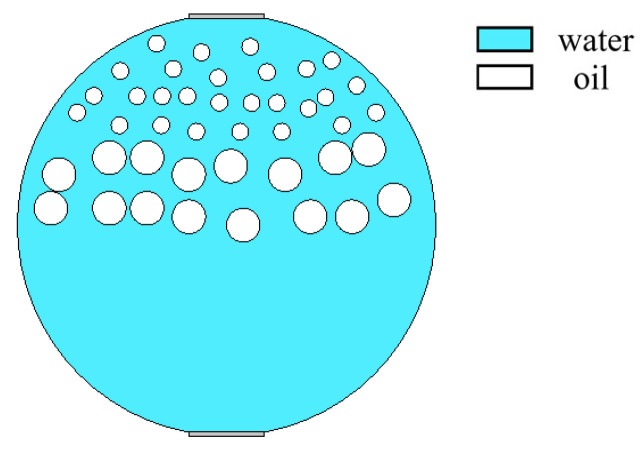
Simulation model of O/W&W flow pattern.

**Figure 9 sensors-22-06128-f009:**
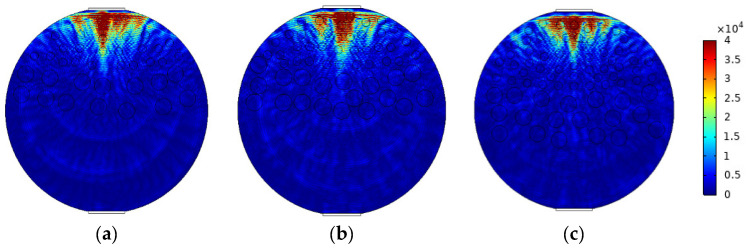
Sound pressure distribution of O/W&W flow pattern: (**a**) oil fraction 10%; (**b**) oil fraction 15%; (**c**) oil fraction 20%.

**Figure 10 sensors-22-06128-f010:**
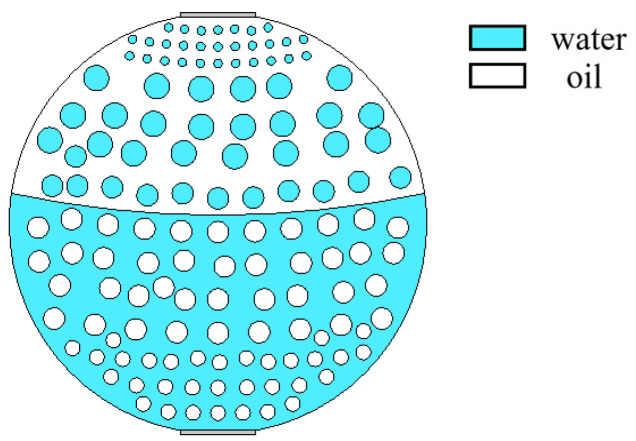
Simulation model of W/O&O/W flow pattern.

**Figure 11 sensors-22-06128-f011:**
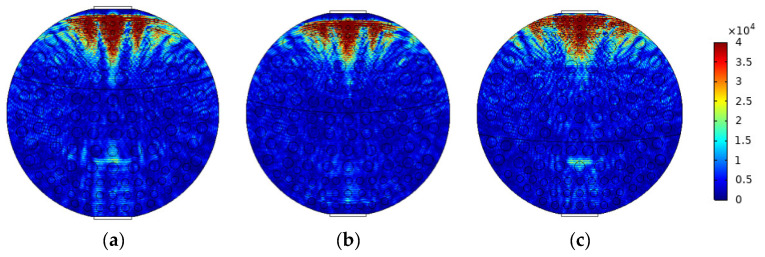
Sound pressure distribution of W/O&O/W flow pattern: (**a**) oil fraction 40%; (**b**) oil fraction 50%; (**c**) oil fraction 60%.

**Figure 12 sensors-22-06128-f012:**
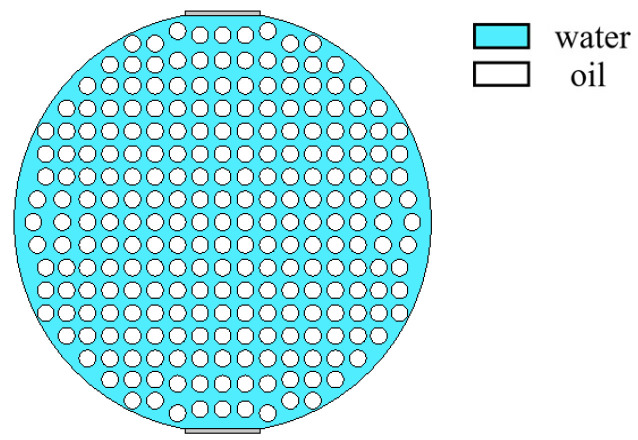
Simulation model of O/W flow pattern.

**Figure 13 sensors-22-06128-f013:**
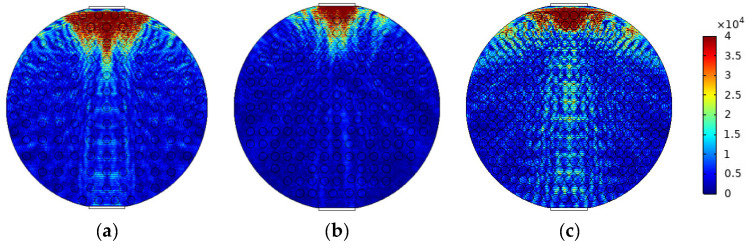
Sound pressure distribution of O/W flow pattern: (**a**) oil fraction 20%; (**b**) oil fraction 40%; (**c**) oil fraction 60%.

**Figure 14 sensors-22-06128-f014:**
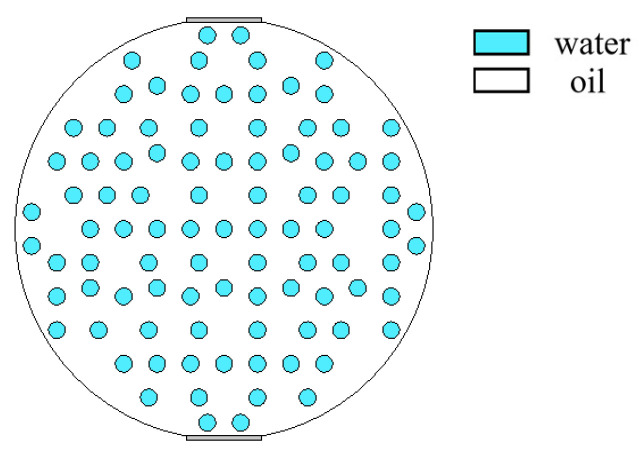
Simulation model of W/O flow pattern.

**Figure 15 sensors-22-06128-f015:**
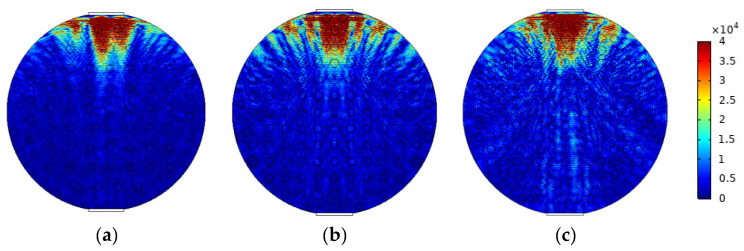
Sound pressure distribution of W/O flow pattern: (**a**) oil fraction 90%; (**b**) oil fraction 80%; (**c**) oil fraction 70%.

**Figure 16 sensors-22-06128-f016:**
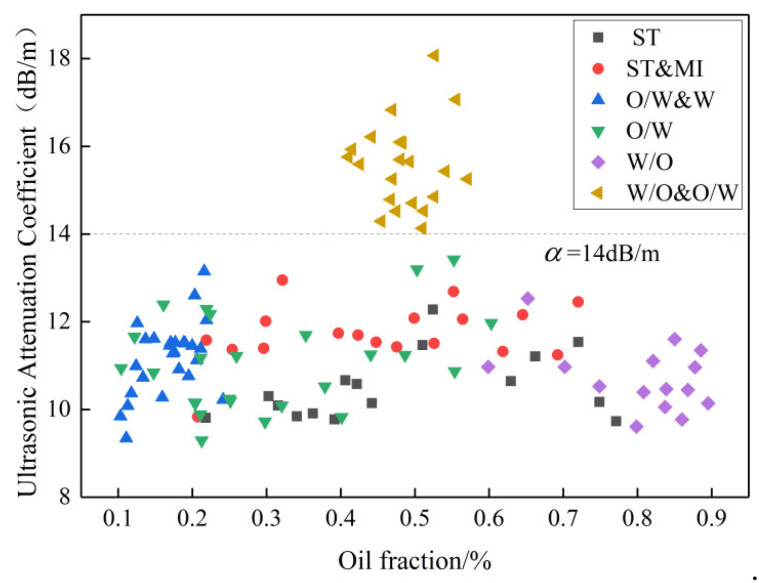
Ultrasonic attenuation coefficient of different flow patterns.

**Figure 17 sensors-22-06128-f017:**
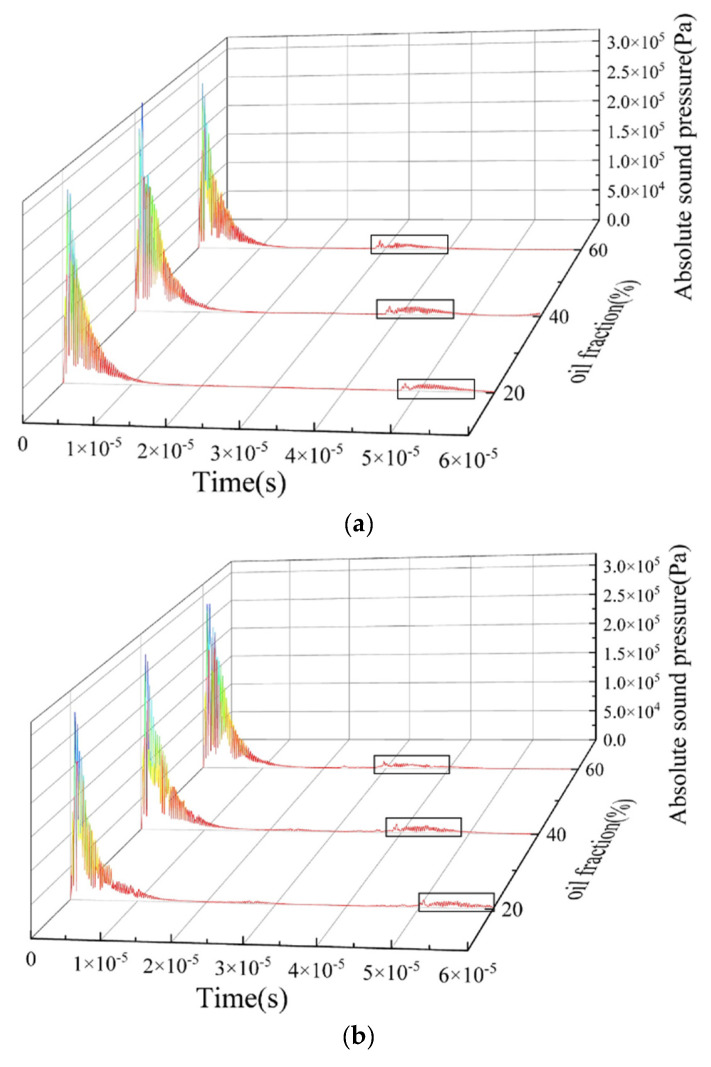
Absolute sound pressure signal of the stratified flow: (**a**) ST flow pattern; (**b**) ST&MI flow pattern; (**c**) O/W&W flow pattern.

**Figure 18 sensors-22-06128-f018:**
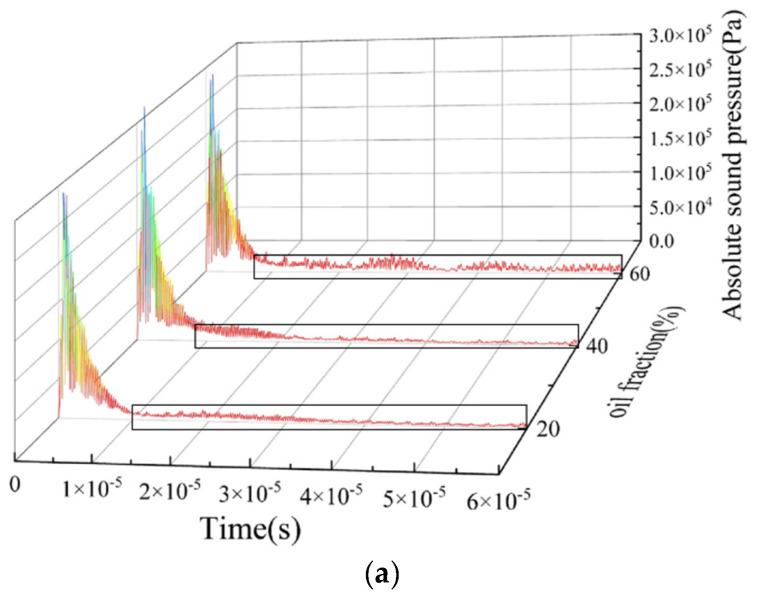
Absolute sound pressure signal of the dispersed flow: (**a**) O/W flow pattern; (**b**) W/O flow pattern.

**Figure 19 sensors-22-06128-f019:**
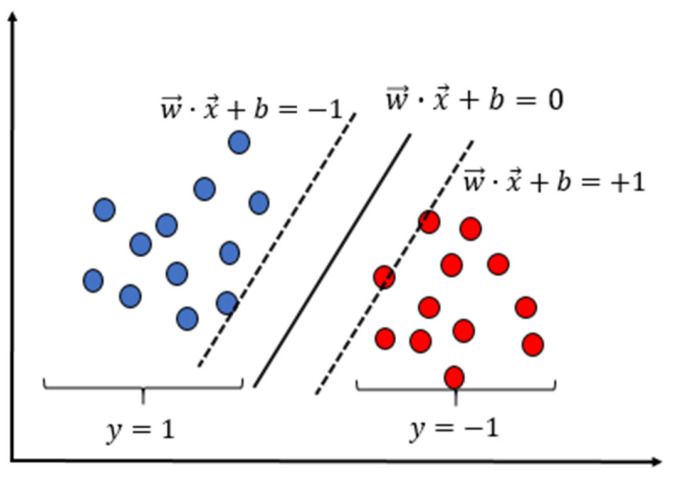
SVM hyper-plane.

**Figure 20 sensors-22-06128-f020:**
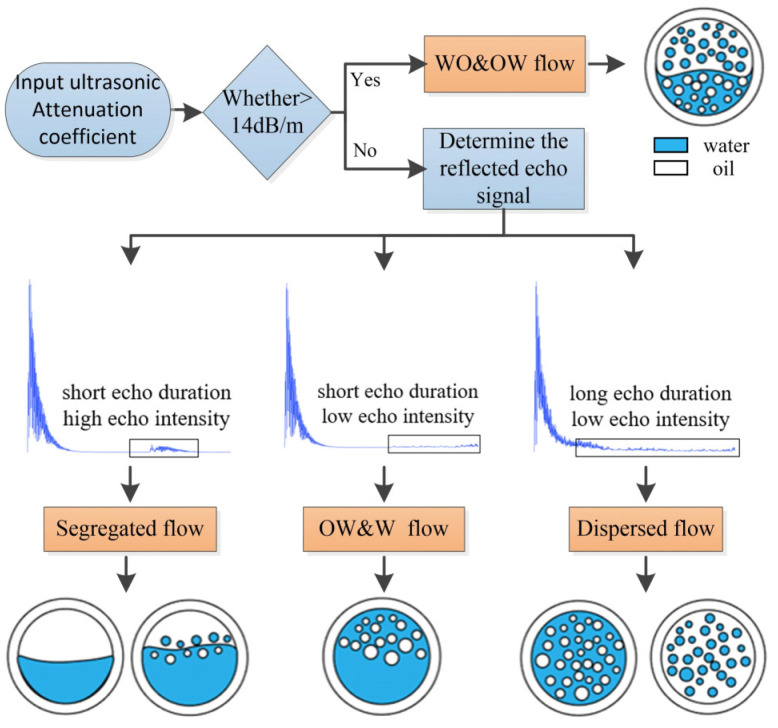
Pattern identification method of oil–water two-phase flow.

**Figure 21 sensors-22-06128-f021:**
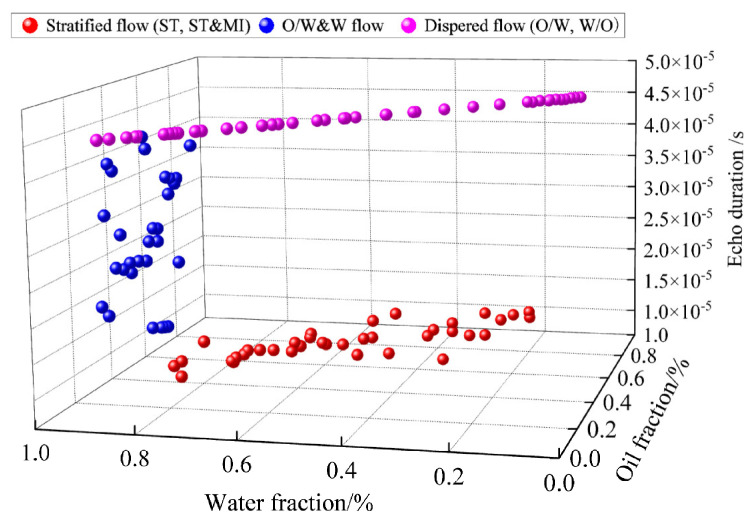
The characteristic distribution of training samples.

**Table 1 sensors-22-06128-t001:** Related physical parameters of two fluids at a 293.15 K environment temperature.

Physical Parameters	Water	Oil
$\mathrm{Sound}\;\mathrm{velocity}/m\cdot s^{-1}$	1448	1420
$\mathrm{Density}//\mathrm{kg}\cdot m^{-3}$	1000	850
Viscosity Pa·s	0.001	0.029
$\mathrm{Thermal}\;\mathrm{conductivity}\;W\cdot m^{-1}\cdot s^{-1}$	0.6	0.2
$\mathrm{Specific}\;\mathrm{heat}\;\mathrm{capacity}/\;J\cdot {\mathrm{kg}}^{-1}\cdot K^{-1}$	4179	2000
$\mathrm{Volume}\;\mathrm{expansion}\;\mathrm{coefficient}/K^{-1}$	$2.1\times 10^{-4}$	$9.0\times 10^{-4}$

**Table 2 sensors-22-06128-t002:** SVM classification results of different kernel functions.

SVM Kernel Function	Computation Time/s	Accuracy/%
Linear	0.20	95.45
Polynomial	0.18	72.73
RBF	0.19	95.45
Sigmoid	0.21	95.45

## Data Availability

Not applicable.

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
