# Peer review of "Flow Pattern Identification of Oil–Water Two-Phase Flow Based on SVM Using Ultrasonic Testing Method"

_sensors, 2022, doi:10.3390/s22166128_

Round 1

Reviewer 1 Report

In this paper, a flow pattern identification method combining ultrasonic transmission attenuation with ultrasonic reflection echo is proposed for oil-water two-phase flow in horizontal pipelines. The topic of the paper is of great interest to readers and the paper is well-organized. I have some minor comments to share with the authors:
1. The misspelled words such as the “An” in Abstract need to be modified.

2. What is the criteria for the selection of the input feature vector of SVM?

3. Why the SVM is selected, and what is the advantage of SVM in flow pattern identification compared with other machine learning algorithms?

4. What needs to be considered when applying this method to the practical experiment in which the fluid is flowing?

5. English writing can be further improved.

Reviewer 2 Report

the paper Flow pattern identification of oil-water two-phase flow based on SVM using ultrasonic testing method written by Su et al. is within the scope of sensors but suffers from critical points :

1- The literature search is weak. There are many related papers in MDPI.
2- Novelty is not clear.

3- Numerical modeling need presentation of equations and boundary condition , grid independency, ...

4- Discussion of technique and results is weak

Reviewer 3 Report

The manuscript is interesting. However, it lacks some parts which should be included in a CFD-focused work.

The governing equations, boundary conditions, and material properties should be included, and a detailed notation afterwards.

In the case of CFD simulation, a mesh independence study should be added.

Stationary or transient simulation was applied?

Model validation (comparison to the measurement) is also missing.

The unit of measurement should be added to the figures.

Are Figure 4, 6, 8, 10 , 12 show the geometry? What was the dimension?

Some smaller remarks:

Page 2. Why is Huang with Capital letters?

Round 2

Reviewer 2 Report

The paper can publish in the current format 

Reviewer 3 Report

The authors addressed my comments, the manuscript can be accepted.